# Dedelayed: Deleting remote inference delay via on-device correction

## Abstract

Remote inference allows lightweight devices to leverage powerful cloud models. However, communication network latency makes predictions stale and unsuitable for real-time tasks. To address this, we introduce *Dedelayed*, a delay-corrective method that mitigates arbitrary remote inference delays, allowing the local device to produce low-latency outputs in real time. Our method employs a lightweight local model that processes the current frame and fuses in features that a heavyweight remote model computes from past frames. On video from the BDD100K driving dataset, Dedelayed improves semantic segmentation accuracy over the stronger of the local-only and remote-only baselines across all realistic communication network delays beyond 33 ms. Without incurring additional delay, it improves accuracy by 6.4 mIoU compared to fully local inference and 9.8 mIoU compared to remote inference, for a round-trip delay of 100 ms. The advantage grows under longer delays and higher-motion scenes, as delay-mitigated split inference sustains accuracy more effectively, providing clear advantages for real-time tasks that must remain aligned with the current world state.

## 1 Introduction

In soft real-time applications—such as cloud gaming or video conferencing—late outputs may be diminished in value, but are still useful. In these applications, we can offload expensive computations to powerful cloud GPUs to save on-device power. As long as the typical latency is low, the loss of utility from latency is outweighed by power savings and extended battery life. However, in hard real-time applications—such as aerial robotic control or obstacle avoidance—late outputs can be catastrophic, and the system must be designed with a guaranteed deadline. Due to the irreducible high-tail latency in wireless communication, hard real-time applications must be equipped with a fully functional local inference pipeline as a fallback in the case that the remote predictions fail to meet the deadline. In this work, we focus on such real-time applications running on resource-constrained devices, relying on inference using video inputs.

In recent years, various approaches to split computing have been proposed to offload computation of expensive image and video models to the cloud to enable the next-generation of robotic, remote sensing, and wearable technology platforms. For real-time streaming video applications, existing approaches still fall into one or more of three common pitfalls. (1) They allocate all on-device power and computation to a single linear inference pipeline, leaving no resources for a local-only fallback. (2) They do not account for the impact of latency on prediction accuracy. (3) They operate on videos with significantly reduced spatiotemporal resolution to manage computational cost, leaving out rich visual details available from modern camera systems.

To address these limitations, we introduce Dedelayed (Fig. 1), a co-inference framework that leverages fresh local information to mitigate the effects of remote inference delay. It consists of a local model and a remote model connected over a communication network, and fuses delayed remote signals into a local pipeline operating on fresh on-device inputs, yielding real-time performance that is never worse than either model in isolation. This enables the use of high-capacity cloud models for delay-sensitive applications while avoiding the pitfalls of prior offloading approaches:

1. **Full integration with local-only fallback model.** No wireless communication channel can offer perfect reliability. For real-time applications with critical deadlines, any remote infer-

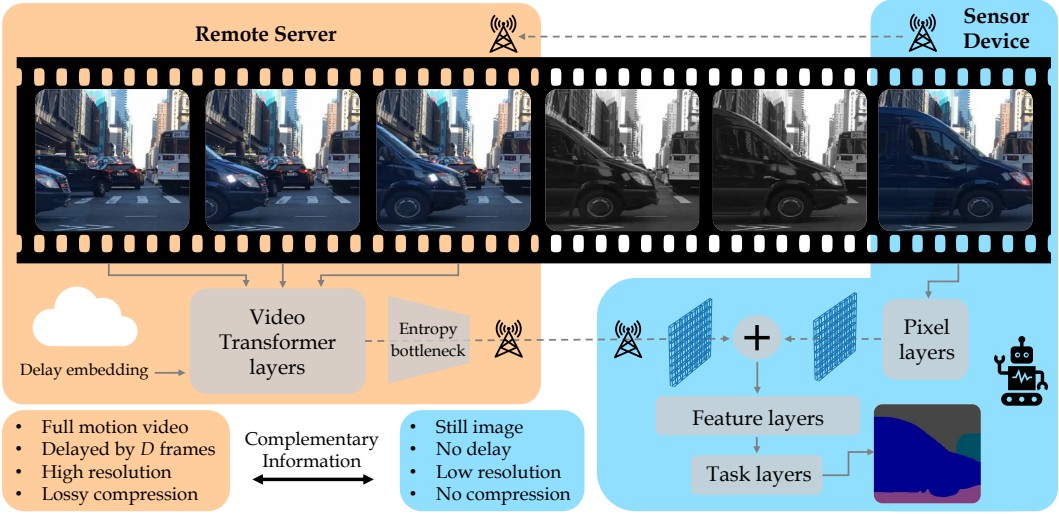

Figure 1: Overview of the Dedelayed real-time inference setup. The lightweight local model and a powerful remote model augment each other's strengths to produce accurate and timely outputs.

ence procedure must be accompanied by a lightweight local fallback model. Instead of two redundant inference pipelines, Dedelayed uses a single path based on a local model that *optionally* incorporates side information from the remote model. We choose a simple method to incorporate this side information—element-wise addition of activation maps—resulting in negligible overhead and well-defined behavior in the absence of remote outputs.

2. **Temporal prediction for latency mitigation.** During supervised training of the remote model component, we simulate a delay of $D$ frames. In other words, the remote model is trained to predict the future. A delay embedding—similar to a position embedding in text or vision transformers—allows the behavior of the remote model to adapt to changes in the channel. As shown in Fig. 2, temporally predictive training is able to capture motion dynamics, which can be used to compensate for latency.

3. **Mixed-resolution inference.** On-device AI video processing at or near the capture resolution and frame rate—typically $> 1$ megapixel and $> 20$ frames per second—is rarely feasible, even with lightweight models. To save resources, real-time computer vision applications often process each frame independently or use small motion updates instead of natively processing a dense 3D pixel volume. Dedelayed is capable of using a mixed-resolution—instead of reducing the resolution for the entire inference pipeline, only the local model resolution is reduced. In parallel, the remote model operates on multiple, high-resolution frames using a 3D transformer. Thus, the remote model can utilize powerful GPUs to model fine details and motion of delayed frames, while the local component can allocate its resources to modeling the current state of objects and the scene, as shown in Fig. 3.

Our contributions are threefold:

1. We provide measurements demonstrating how higher degrees of latency hurt the accuracy of dense visual prediction for semantic segmentation of driving scenes.

2. We introduce Dedelayed, a split computing framework for video inputs that integrates the output of a future-predicting remote model with the current output of a local model.

3. Using Dedelayed, we create a video segmentation system for urban driving scenes that outperforms any existing local or remote inference solution, while avoiding the pitfalls that limit the practicality of previous approaches.

For experimental validation, we demonstrate a simple Dedelayed system with an addition-based fusion on off-the-shelf models with minimal architectural changes. This makes it easy to enhance existing pipelines, deploy in practice, and extend to other real-time methods.

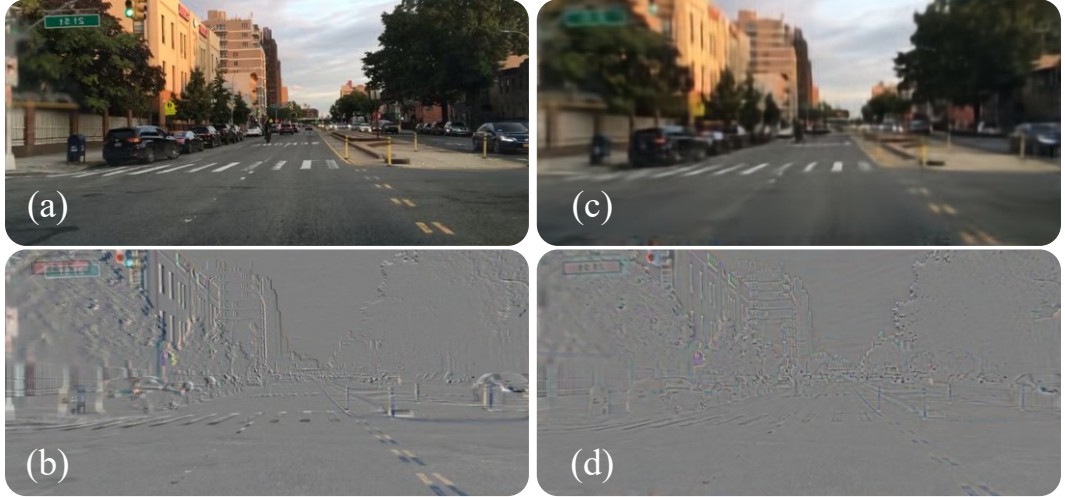

Figure 2: To demonstrate the effect of temporally predictive training, we train a 3D transformer to predict the next frame with an MSE loss on pixels. (a) shows the original video frame. (b) shows the difference between (a) and a future frame, with objects such as the traffic sign and road markings in different locations. (c) shows the pixel predictions of the 3D transformer. (d) shows the difference from the true future frame. While the predictive model cannot predict high-frequency details, it is able to accurately model the motion of objects, signs, and road markings.

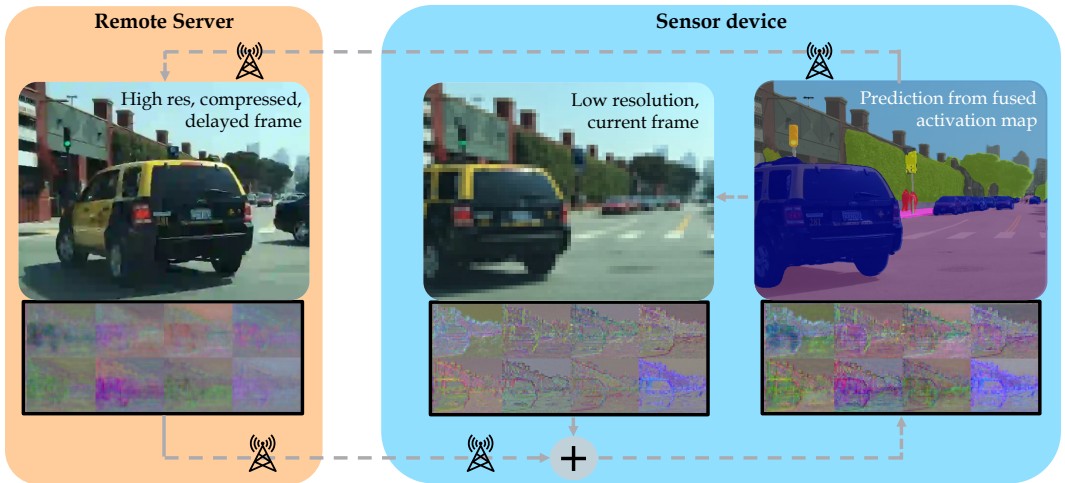

Figure 3: Example of activation maps from local and remote model components. The remote server uses the higher level of video detail to accurately distinguish and classify objects. The local model provides exact position adjustments based on the current frame. When making predictions from the combined activation map, small details that would be impossible to make out at low resolution (e.g., the distant pedestrians, labeled red) are accurately classified and localized.

## 2 BACKGROUND

In the human visual system, the optic nerve can only transmit a small fraction of the information received by the retina (Kelly, 1962). Barlow's efficient coding hypothesis (Barlow et al., 1961) posits that compression is the primary role of early processing; once this compressed representation is received in deeper layers of the visual cortex, more metabolically intense processing can occur. In the predictive coding model (Rao & Ballard, 1999), this processing is driven by feedback mechanisms that minimize a temporally predictive error signal to create a perceptual model that is consistent with sensory inputs.

Machines equipped with digital video sensors—which are at the center of ongoing innovation in robotics (Kim et al., 2024; O'Neill et al., 2024), remote sensing (Szwarcman et al., 2024; Khani et al., 2021), and wearable technology (Grauman et al., 2022; 2024)—share similar constraints. The throughput and power efficiency of ingesting pixels on the sensor device (e.g., a battery-powered robot) are extremely high—typically tens or hundreds of megapixels per watt-second (Engel et al., 2023). However, moderately sized DNNs can only process visual data at about one megapixel per watt-second (Cai et al., 2023). For more advanced video AI based on autoregressive modeling (Agarwal et al., 2025) or temporal prediction (Assran et al., 2025), the efficiency may be as low as 500 pixels per watt-second. Instead of on-device processing, power constraints can be circumvented by compressing and transmitting video streams to cloud GPU datacenters supported by a 100-megawatt power infrastructure (Goldberg & Kehoe, 2013).

However, fully remote processing is challenging for certain real-time applications (e.g., collision avoidance) due to unreliability in network and cloud infrastructure (Chen et al., 2024; 2025). Thus, delivering predictions by a guaranteed deadline requires a fallback procedure independent of the remote server. In many systems (e.g., autonomous motor vehicles) the limited accuracy and reliability of lightweight local models warrant a human operator as the fallback (Committee, 2021), preventing full automation.

## 3 RELATED WORK

Prior work has extensively explored lightweight architectures (e.g., EfficientViT (Cai et al., 2023), MobileNetV4 (Qin et al., 2024)) that minimize computation to achieve real-time on-device performance. These approaches deliver low latency but are constrained by device power and compute. When devices are too limited, the common alternative is fully remote inference, which offloads computation to servers, but is highly susceptible to network latency.

Split-computing approaches largely focus on distributing workloads rather than optimizing for strict real-time operation. Next-generation compression standards such as MPEG AI (ISO/IEC, 2025) and JPEG AI (Ascenso et al., 2023) target bandwidth reduction via task-specific compression, significantly lowering transmission costs with reasonable compute overhead. However, even they lack explicit mechanisms to anticipate or compensate for network delay, leading to stale predictions misaligned with the current world state.

Other efforts related to our work have also been explored. Clockwork Convnets (Shelhamer et al., 2016) reuse stale features to reduce inference latency, but they offer limited temporal reasoning and operate on a single device. Accel (Jain et al., 2019) warps heavy-model features forward with optical flow and corrects them with a lightweight model, but is also not intended for across-network operation. Adaptive Model Streaming (Khani et al., 2021) streams weight updates from a server to keep a local model fresh, focusing on model adaptation rather than directly mitigating per-frame staleness from communication latency. Though less known, Knowledge Boosting (Srinivas et al., 2024) is very closely related to our work. Like us, it fuses delayed remote features with a small on-device model, but it assumes a fixed delay. We generalize to longer and variable latencies by conditioning on a tunable delay while keeping the design simple and reusable.

## 4 METHOD

### 4.1 DELAY MITIGATION FRAMEWORK

Dedelayed introduces a general framework that improves the accuracy and robustness of real-time inference on resource-constrained sensor devices. It does so by combining the strengths of both local inference and remote inference, while mitigating their weaknesses. The local model has access to the latest sensor data, and yet lacks the computational capability needed to produce accurate outputs. The remote model provides accurate outputs, and yet delivers them with delay. By careful combination of both subsystems, Dedelayed is able to provide bounded performance guarantees—it is never worse than either local inference or remote inference independently. As we will demonstrate later, we are able to glue together the two subsystems in a way that is simple yet effective.

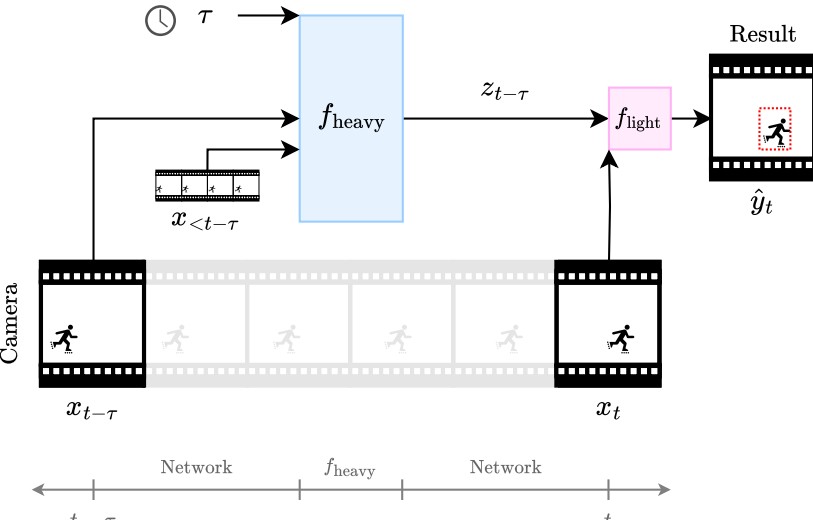

Figure 4: Time progresses left to right. The client-side camera produces video frames, which are sent across a communication network to the server. The server runs a heavyweight model using the latest video frame $x_{t-\tau}$ that it receives, in addition to a context of previously received video frames $x_{<t-\tau}$, as well as the measured delay $\tau$. This produces an output $z_{t-\tau}$ that the server sends to the client. The client pairs the latest received response $z_{t-\tau}$ with a freshly produced video frame $x_t$, and runs these inputs through a lightweight model. This finally produces a timely result $\hat{y}_t$ that can be used in real-time delay-sensitive applications.

Dedelayed addresses the problem of stale predictions from powerful remote models by integrating them with a lightweight, on-device model. The core idea is to leverage the high-quality features from a heavyweight remote model, despite their inherent delay, by explicitly conditioning them on the measured latency and fusing them early with live information from a local model. This ensures that the final predictions are both accurate and timely.

Dedelayed can be formulated in simple mathematical terms as follows. Given a fresh input frame $x_t$ at current time $t$, the final prediction $\hat{y}_t$ is computed using a lightweight local model, $f_{\text{light}}$, which processes $x_t$ along with time-delayed features $z_{t-\tau}$ from a heavyweight remote model, $f_{\text{heavy}}$. To produce powerful predictive features, the remote model is conditioned on the delay $\tau$, and processes a short clip of past frames $x_{\leq t-\tau}$ ending at time $t-\tau$. This is expressed by the following equations:

$$z_{t-\tau} = f_{\text{heavy}}(\tau,\, x_{\leq t-\tau}) \tag{1}$$

$$\hat{y}_t = f_{\text{light}}(x_t,\, z_{t-\tau}) \tag{2}$$

For clarity, the notation is summarized in Table 1. Fig. 4 presents a system diagram that demonstrates the fundamental principle we describe, and shows how information propagates through the various subsystems as time progresses.

Table 1: Notation.

| Symbol | Meaning |
|---|---|
| $x_t$ | Input frame at current time $t$ |
| $x_{\leq t-\tau}$ | Input frames up to time $t-\tau$ |
| $\hat{y}_t$ | Prediction for time $t$ |
| $z_{t-\tau}$ | Features outputted by heavyweight model run at time $t-\tau$ |
| $\tau$ | Delay in time between the old and current frame at time $t$ |
| $D$ | Delay in frames between the old and current frame at time $t$ |
| $f_{\text{light}}$ | Lightweight model run at time $t$ |
| $f_{\text{heavy}}$ | Heavyweight model run at time $t-\tau$ |

The entire Dedelayed system is trained end-to-end to minimize a task-specific loss function, $\mathcal{L}_{\text{task}}$, evaluated against the ground truth $y_t$ for the current frame.

$$\mathcal{L}_{\text{task}} = \ell(\hat{y}_t, y_t)$$

For semantic segmentation, $\ell$ is typically the cross-entropy loss. The objective is to produce predictions $\hat{y}_t$ that are accurate at time $t$ on the local device.

In the next section, we detail how we designed a specific implementation to test the Dedelayed framework in action.

### 4.2 Design and Implementation

Dedelayed consists of a lightweight on-device model and a predictive remote model connected over a communication network. The remote side computes delay-conditioned features from past inputs and returns them to the device, where they are fused early into the local model running on the current input. To provide a guarantee on baseline performance, these components can first be trained to work independently and later fused and trained jointly.

**System overview**   Information propagates through our system as follows:

1. The local device transmits input frames to the remote via the uplink.

2. Each incoming frame is independently encoded into features using a pretrained 2D ViT backbone on the remote. We maintain a context window of the $K$ most recent features.

3. The $K$ per-frame features are concatenated along the temporal axis, and a learned delay embedding conditioned on the measured delay $\tau$ is added.

4. A 3D ViT encoder followed by learned pooling (MLP–pool–MLP) produces delay-conditioned remote features $z_{t-\tau}$, which are sent back to the device via the downlink.

5. The lightweight local model runs on a fresh input $x_t$, and fuses in the remote features $z_{t-\tau}$.

6. The local model finishes decoding the fused representation and outputs labels $\hat{y}_t$.

**Remote predictive module.**   Figure Fig. 5 visualizes the remote component. The remote model processes a fixed context of $K$ past frames ending at $t - \tau$. Each frame is encoded independently with a 2D ViT—we use EfficientViT-L1 with an effective $8 \times 8$ patch. The per-frame features are concatenated along the temporal axis and spatially merged into larger $16 \times 16$ patches to keep the sequence length comparable. (When $K = 4$, the sequence length is identical.) A learned delay embedding determined by the delay $\tau$ is added, and the result is then processed by a 3D ViT followed by learned pooling (MLP–pool–MLP) to produce delay-conditioned features $z_{t-\tau}$. We pretrain the remote predictive module by attaching a task head to the 3D ViT backbone and training it to predict the target labels $y_t$ from inputs up to time $t - \tau$.

**Local model and fusion.**   The lightweight local model processes the fresh input $x_t$. We compute first-stage features $h = \texttt{T1}(x_t)$ and perform early fusion by element-wise addition with the delayed remote features, $h' = h + z_{t-\tau}$. Both tensors have shape $96 \times H/8 \times W/8$, so no projection or resizing is required. The fused features $h'$ are then processed by the remaining local blocks and decoded to the final output $\hat{y}_t$. If $z_{t-\tau}$ is unavailable, the local model falls back to $h' = h$ and proceeds unchanged.

## 5 Experiments

We focus our evaluation of Dedelayed on the task of real-time semantic segmentation of driving scenes, a domain where timely and accurate perception is critical for applications such as autonomous driving and robotics. We demonstrate that delay-aware feature fusion can mitigate remote inference latency, sustaining accuracy even when remote predictions are delayed by long communication network latencies.

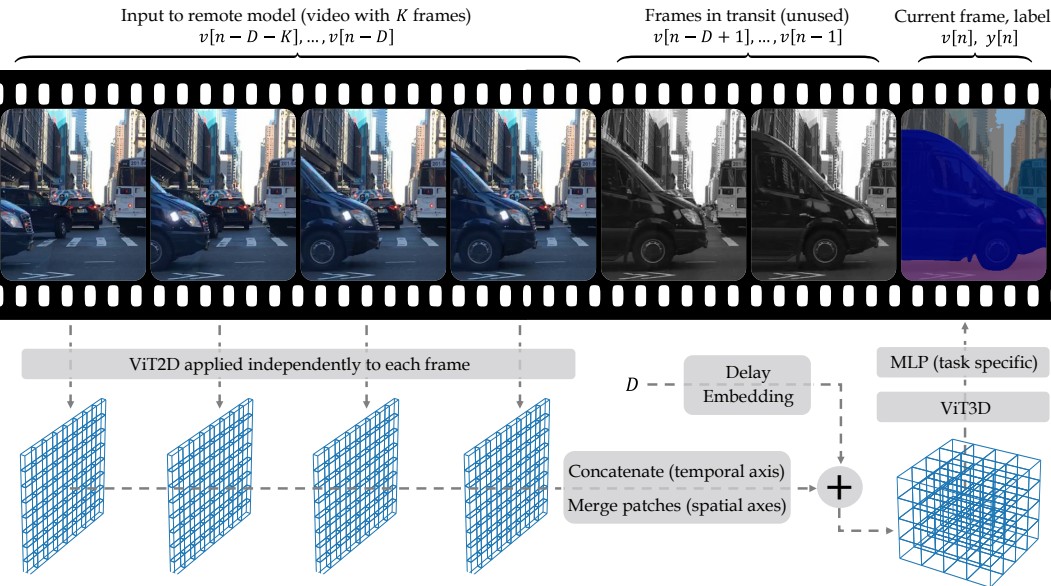

Figure 5: Overview of the remote video predictive model, trained to predict the label at index $n$ from a $K$-frame context ending at $n - D$.

### EXPERIMENTAL SETUP

Our experiments utilize the BDD100K video dataset (Yu et al., 2020), containing video of driving scenes at 30 frames per second (fps). Since the dataset does not provide dense segmentation labels for all video frames, we generated pseudo-labels using a pretrained EoMT (Kerssies et al., 2025) model, ignoring pixels with low confidence. We use a subset of 19 labels from Cityscapes.

Our evaluation covers a range of realistic network delays, from 0 to 5 frames, corresponding to 0 to 165 ms. This range is representative of typical round-trip latencies and is sufficient to demonstrate the degradation of conventional remote inference and the resilience of Dedelayed. At training time, the delay $\tau$ was sampled per batch from a uniform distribution over this range. This delay, together with the relevant frames from the video, was fed into the remote model. To replicate real-world usage, we applied compression to the uplink video streams. We chose the resolution and frame rate to fit within reasonable uplink capacity using the lossy WebP image codec at quality 85.

### TRAINING DETAILS

We adopt a multi-stage training strategy, as detailed in Table 2. The remote and local models are first trained individually and then later combined. Each model is pretrained on the large-scale ImageNet dataset (Russakovsky et al., 2015) for classification, then on the image segmentation task on Cityscapes (Cordts et al., 2016), before being fine-tuned on the smaller BDD100K driving dataset. We specifically train the remote model to have predictive capability by supplying it with a delay-aware (DA) objective: to predict the labels of future frames, conditioned on a tunable delay. By training in stages, we are able to provide guarantees on baseline remote and local model performance. Finally, the remote and local models are glued together, and the full system is jointly fine-tuned on the delay-aware task of streaming semantic segmentation on video.

We train using cross-entropy loss, the Adan (Xie et al., 2024) optimizer, a trapezoidal cosine learning rate schedule, gradient clipping, and selectively applying discriminative fine-tuning or layer-wise learning rate decay (LLRD) (Howard & Ruder, 2018).

Table 2: Training stages for remote, local, and fusion models.

| Model | Stage | Obj. | Data | Epochs | Res. | Freeze |
|---|---|---|---|---|---|---|
| **Remote (image only)** | | | | | | |
| SegFormer-B5 | pre- | DU | IN1K, CS | – | – | – |
| SegFormer-B5 | 1 | DU | BDD | 15 | 496 | – |
| **Remote (video-predictive)** | | | | | | |
| EfficientViT-Seg-L1 | pre- | DU | IN1K, CS | – | – | – |
| EfficientViT-Seg-L1+ViT3D | 1 | DA | BDD | 10 | 496 | img-bkbn |
| EfficientViT-Seg-L1+ViT3D | 2 | DA | BDD | 10 | 496 | – |
| **Local (image only)** | | | | | | |
| MSTransformer2D | 1 | DU | IN1K | 320 | 224 | – |
| MSTransformer2D | 2 | DU | CS | 80 | 336 | bkbn + proj |
| MSTransformer2D | 3 | DU | CS | 80 | 336 | LLRD 0.9 |
| MSTransformer2D | 4 | DU | BDD | 15 | 496 | LLRD 0.9 |
| **Fusion (local image + remote video-predictive)** | | | | | | |
| MSTransformer2D + EfficientViT-Seg-L1+ViT3D | 1 | DA | BDD | 10 | 480/720 | remote img-bkbn + 3D |

Stage: pre- = pretrained (external source).
Obj.: DA = delay-aware objective; DU = delay-unaware objective.
Data: IN1K = ImageNet-1K; CS = Cityscapes; BDD = Berkeley DeepDrive 100K.
Freeze: bkbn + proj = backbone & MS-projections frozen; remote img-bkbn + 3D = remote image backbone and remote 3D encoder frozen; LLRD = layer-wise learning rate decay.

# 6 RESULTS

We compare how various inference systems perform under the effect of communication network latency. Fig. 6 visualizes the various local-only, remote-only, and fused local+remote methods from different stages of our training. Each baseline is effectively an ablation of our final system.

- **Local image** and **Remote image** inference setups process individual frames in the conventional way, though the remote is susceptible to communication network delay.
- **Remote video** has access to past frames of context, but only predicts labels for its present view, and thus fares no better than "remote image".
- **Remote predictive** is fed a tunable delay and sustains accuracy by predicting the future.
- **Local + remote predictive** represents a Dedelayed system, and is thus able to further sustain accuracy by merging the remote predictive features with fresh local features.

While the above results demonstrate the dominance that Dedelayed is capable of, one should also account for the impact of local inference delay. We show this in Fig. 7 by assessing accuracy versus total latency. For local inference delays of $\leq 8$ ms, the "local + remote predictive" method is consistently better across all network round-trip delays in terms of *both* accuracy and total latency.

# 7 CONCLUSION

Dedelayed addresses a central challenge in real-time systems that rely on remote computation: prediction staleness induced by network delay. It mitigates remote inference delay by elevating delay to a first-class variable, conditioning the remote model via a learnable delay embedding, and fusing remote features with fresh local features. Across realistic network conditions, Dedelayed surpasses strong local-only and remote-only baselines, with a particular advantage for longer latencies and high-motion content. As a foundational framework, Dedelayed applies to a wide range of real-time problem domains, enabling intelligent systems that are not only accurate but also truly timely and dependable in dynamic environments. Future work includes studying variable and stochastic delay distributions, high-motion data, lighter local models, and local future prediction.

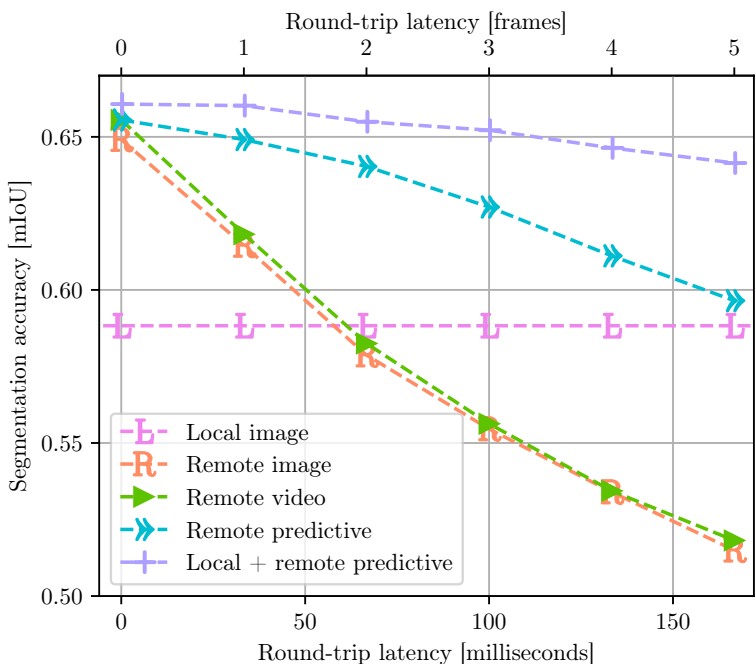

Figure 6: Segmentation accuracy (mIoU) versus round-trip latency (milliseconds or frames).

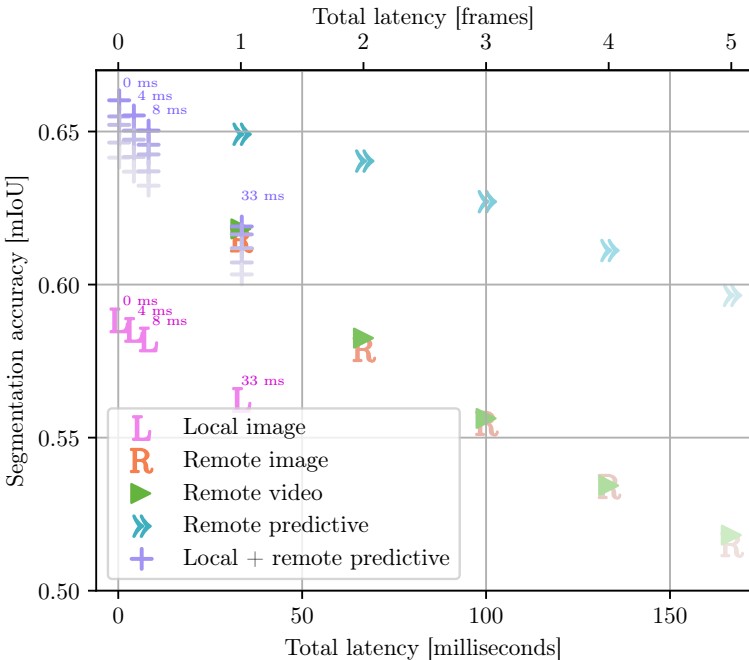

Figure 7: Segmentation accuracy (mIoU) versus total latency (milliseconds or frames) for selected local model delays. Points are faded as round-trip latency increases. 4 and 8 ms were interpolated.

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

# A APPENDIX

FREQUENTLY ASKED QUESTIONS (FAQ) FOR REVIEWERS

Our experiments demonstrate the value of the proposed system and validate our central hypothesis: a properly designed Dedelayed system—when composed of the baseline components—should never perform worse than either baseline alone. Nonetheless, we provide clarifications in advance addressing anticipated reviewer questions about our evaluation procedure.

---

*Q: Why not use a video segmentation dataset like DAVIS?*

*Why focus on segmentation?*

*Why use the Cityscapes taxonomy if you are using BDD100K?*

In the computer vision literature, it is common to include evaluations on image segmentation datasets (e.g., Cityscapes or ADE20K) when proposing new training techniques or model architectures. For example, foundation models like DINOv3 (Siméoni et al., 2025) include evaluations on these datasets. By evaluating semantic segmentation performance using the Cityscapes taxonomy, we can more easily compare our proposed method to a plethora of other model architectures and training procedures. Additionally, we can use the pretrained weights of Cityscapes segmentation models from independent researchers exactly as provided or with minimal fine-tuning. This allows us to provide a strong set of baselines to compare against. Semantic segmentation also involves simpler architectures compared to the interactive or semi-supervised objectives often used for video segmentation datasets like DAVIS.

---

*Q: Why not compare against video segmentation models (e.g., SAM 2)?*

Our models are evaluated on the per-frame *image segmentation* task. This may seem surprising since we evaluate on video sequences. However, we only use video sequences to simulate delay and to provide temporal context to improve predictive performance. The local model takes a single frame $x_t$ and predicts the corresponding label $y_t$ for that single frame.

In contrast, interactive video segmentation models by default assume offline access to the full video sequence $\{x_1, \ldots, x_N\}$ in advance. This is not ideal for real-time applications, which must process frames incrementally frame-by-frame under strict latency constraints. Many popular video segmentation models are not ideal for real-time applications for a further reason still: they involve two passes. In the first pass, an initial set of segmentation masks are generated based on the first frame. In the second pass, segmentation masks from the first frame are adjusted based on subsequent frames. The two passes also involve different amounts of processing time, adding significant operational complexity for real-time applications.

Additionally, to meet the requirements of on-device processing, most image and video segmentation models dramatically reduce the resolution (e.g., from 1080p or 720p to $448^2$ or even $224^2$). Leveraging multi-frame context generally increases computation, often requiring even lower resolution to meet real-time deadlines. While this trade-off can help in some settings, the evidence is less compelling than the widely observed gains from higher resolution.

Finally, we argue that any model able to incorporate multiple frames of context while only adding a negligible amount of computation compared to a single-frame model—e.g., by using mask propagation like SAM 2—will be prone to other failure modes. For example, large motions leading to blur (e.g., a 90° camera turn) or transient poor lighting conditions (e.g., brief direct or reflected sunlight) could cause catastrophic results on keyframe, leading to poor performance on subsequent frames. On the other hand, independently segmenting each frame limits the effects of these transients, allowing the output quality to be restored as soon as the input quality improves. Thus, a per-frame semantic segmentation model is a more versatile and appropriate objective to use in evaluating our proposed system.

---

*Q: Why use pseudolabels?*

*Why not use the human annotated segmentation masks provided in the BDD100K dataset for training and evaluation?*

*Have you verified that the pseudolabels used for evaluation are reliable?*

To our knowledge, no major video dataset contains reliable and spatiotemporally dense human annotations for a standard supervised learning task (e.g., Cityscapes or ADE20K segmentation). For BDD100K, the annotations are not temporally dense and they are rarely used by other researchers. We have manually verified dozens of pseudolabel masks generated for the evaluation pipeline. In our subjective evaluation, the pseudolabels have fewer errors than the human annotations in datasets like ADE20K or Cityscapes, even in poor lighting or when noticeable motion blur is present.

---

*Q: Why use 500 randomly selected videos instead of the entire BDD100K validation set?*

We choose 500 videos to match the size of the Cityscapes evaluation dataset, while allowing more rapid evaluation and reducing the cost of generating the high quality pseudolabels using DepthAnything+Mask2Former.

## A.1 CODE

The fused "local + remote predictive" model is defined below:

```python
class FusedModel(nn.Module):
    def __init__(self, cls_classes=1000, seg_classes=19):
        super().__init__()
        self.local_model = MSTransformer2D(cls_classes, seg_classes)
        self.remote_model = EfficientViTSeg3D()
        self.mlp_pre_pool = PrepoolBlock()
        self.mlp_post_pool = PostpoolBlock()

    def forward(self, x_local, x_remote, delay):
        # Local feature extraction:
        h = self.local_model.T1(x_local)
        _, _, H, W = h.shape

        # Remote feature extraction and pooling:
        z = x_remote
        z = self.remote_model.forward_features(z, delay)
        z = einops.rearrange(z, "b c f h w -> b (c f) h w", f=4)
        z = self.mlp_pre_pool(z)
        z = F.adaptive_avg_pool2d(z, output_size=(H, W))
        z = self.mlp_post_pool(z)

        # Local and remote feature fusion:
        h = h + z

        # Remaining local model:
        y = h
        h = self.local_model.T2(h)
        y = y + self.local_model.P2(h)
        h = self.local_model.T3(h)
        y = y + self.local_model.P3(h)
        y = self.local_model.seg_head(y)

        return y

class EfficientViTSeg3D(nn.Module):
    def __init__(self, name="efficientvit-seg-l1-cityscapes", seg_classes=19):
        super().__init__()
        self.image_model = create_efficientvit_seg_model(name)
        self.delay_embedding = DelayEmbedding()
        self.vit3d = nn.Sequential(*[
            VitBlock3D(in_channels=256, head_dim=32, expand_ratio=4)
            for _ in range(12)
        ])
        self.head = RemotePredictiveHead(seg_classes=seg_classes)

    def pool(self, x):
        x_pool = F.adaptive_avg_pool3d(x, output_size=(4, x.shape[-2:]))
        return einops.rearrange(x_pool, "b c f h w -> b (c f) h w", f=4)

    def forward_features(self, x, delay):
        video_embedding = self.image_model.backbone(x)  # shape: (B, C, F, H, W)
        delay_embedding = self.delay_embedding(delay, video_embedding.shape)
        return self.vit3d(video_embedding + delay_embedding)

    # For pretraining remote predictive model:
    def forward(self, x_remote, delay):
        return self.head(self.pool(self.forward_features(x_remote, delay)))

class MSTransformer2D(nn.Module):
    """Any 2D image segmentation model. We use one with T1, T2, T3 blocks."""
```

To train or evaluate the models, simply feed in frames separated by the appropriate delay and compute cross-entropy loss or mIoU in the typical fashion.

## A.2 Effect of delay jitter

We evaluate how our model performs under delay jitter, i.e., when the delay varies for each frame.

Our model is trained only to maximize accuracy for a fixed, tunable delay input and is not explicitly optimized for jitter. Nonetheless, temporal structure in the data allows it to retain accuracy even when the delay input differs from the observed delay. Fig. 8 characterizes this, showing performance under different observed delays when the model is force-fed a possibly inaccurate delay as input. Accuracy peaks when the delay input matches the observed delay, and the drop is less sharp when the observed delay exceeds the delay input. The latter might be attributed to selection bias—if the predicted features are relevant for a future frame, they often remain useful for subsequent frames.

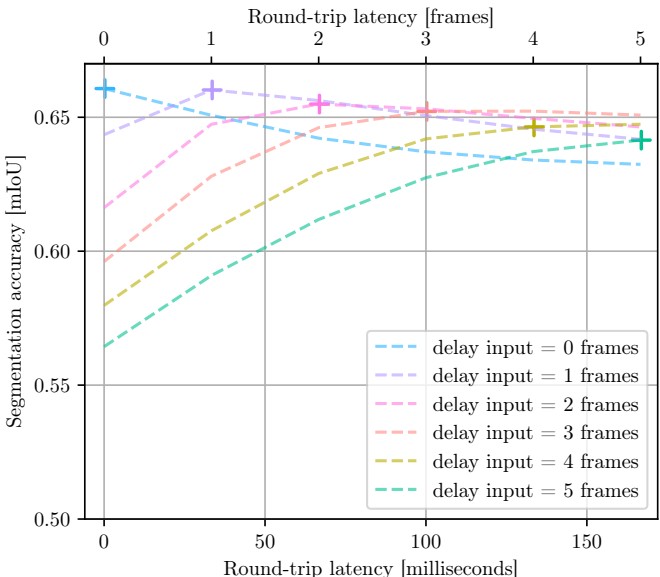

Figure 8: Segmentation accuracy (mIoU) versus round-trip latency (milliseconds or frames). Here, the remote model's delay input has been force-fed a specific value.

In Fig. 9, we report the above results in matrix form. Although our model was not explicitly trained for mismatched delays or delays beyond 5 frames, it continues to perform well under these conditions. This matrix can also be used to estimate the performance under delay jitter. We model the observed delay as $\tau_{\text{obs}} \sim \mathcal{N}(\mu = \tau_{\text{in}}, \sigma^2)$, centered at the delay input to the model, $\tau_{\text{in}}$. The expected accuracy is obtained by taking a weighted sum over mIoU values for each observed delay, using a discretely binned normal probability mass function, with out-of-bounds mass assigned to the boundary bins. The resulting performance is shown in Fig. 10 for various values of $\sigma$. In many networks, $\sigma = 5$ ms and $\sigma = 15$ ms correspond to relatively high jitter. Yet, even for higher values of $\sigma$, our method's performance maintains its advantage, even assuming optimistically that the competing baselines experience no jitter. This shows that our method performs stably in realistic networks. For comparison, traditional remote inference loses 3.4% mIoU within the first frame of delay, a drop far larger than the degradation our method incurs with the corresponding $\sigma$.

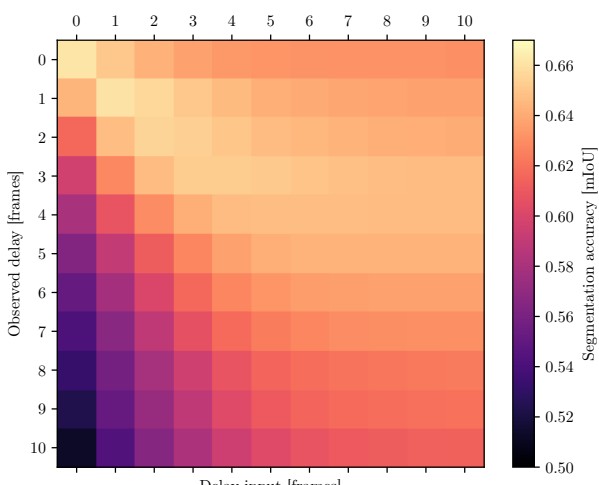

Figure 9: Segmentation accuracy (mIoU) over observed delay and model delay input.

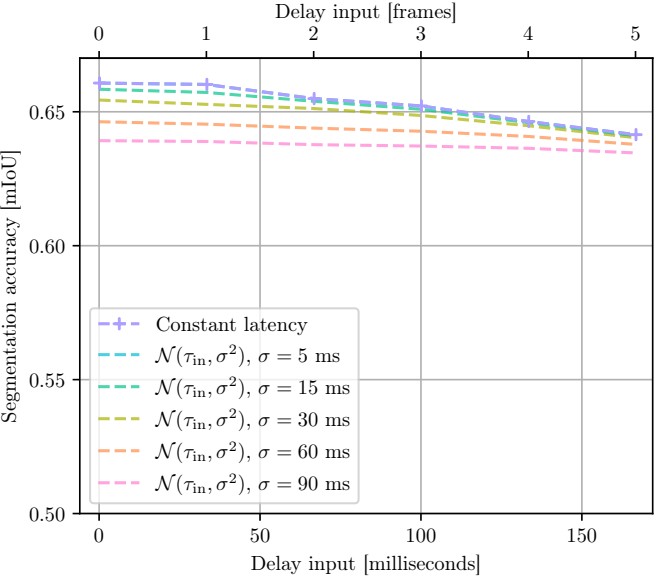

Figure 10: Segmentation accuracy (mIoU) versus round-trip latency (milliseconds or frames). Latency jitter is modeled as a normal distribution.

### A.3 LOCAL INPUT RESOLUTION

We evaluate on various local input resolutions. Our remote-assisted local model is able to operate at far lower resolutions without losing accuracy.

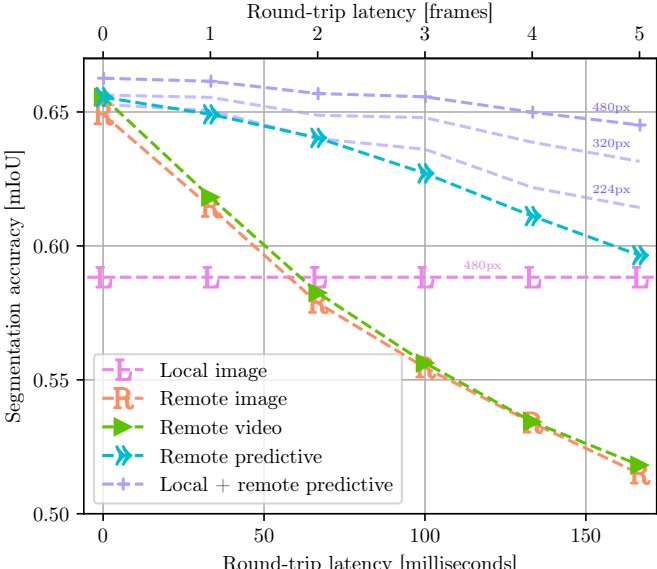

Figure 11: Segmentation accuracy (mIoU) versus round-trip latency (milliseconds or frames). Further fine-tuned and evaluated on various local input resolutions.

