# OpenReview forum: "Dedelayed: Deleting remote inference delay via on-device correction"
_ICLR.cc/2026/Conference — ICLR 2026 Conference Withdrawn Submission_

### Official Review · Reviewer_kyRZ · 2025-10-29

**Soundness:** 3
**Presentation:** 2
**Contribution:** 3
**Rating:** 6
**Confidence:** 4

**Summary:**

This paper proposes Dedelayed,  a split-inference framework that mitigates remote inference latency in real-time applications. The approach combines a lightweight local model that processes current sensor frames with a remote predictive model that performs delay-aware future prediction from past frames. The remote model produces features conditioned on a learnable delay embedding, which are fused into the local model via simple element-wise addition. Experiments on the BDD100K driving dataset demonstrate consistent improvements in semantic segmentation accuracy over both local-only and remote-only baselines across a wide range of network latencies. The method also shows robustness to delay jitter and reduced input resolution. Overall, the paper introduces a conceptually simple yet effective framework to correct stale predictions in cloud-assisted real-time perception.

**Strengths:**

**Originality:** Introduces the idea of “delay embedding” and future-predictive remote inference for real-time correction—conceptually fresh and elegant.

**Practicality:** The fusion method (element-wise addition) is simple, hardware-friendly, and easy to integrate into existing pipelines.

**Empirical validation:** Covers multiple factors—latency magnitude, jitter, input resolution—and demonstrates consistent accuracy gains.

**Relevance:** Addresses an increasingly important bottleneck in real-time perception and robotics: network-induced staleness in remote inference.

**Weaknesses:**

1. **Lack of theoretical backing:** No analytical guarantee or formal proof for the “never worse than baseline” claim.
2. **Data quality concern:** Uses pseudo-labels instead of human annotations, potentially biasing evaluation.

3. **Incomplete technical clarity:** The delay embedding mechanism and fusion location are insufficiently described.

4. **Presentation issues:** Some figures are duplicated or mislabeled (e.g., Fig. 11), and notation inconsistencies reduce clarity.

**Questions:**

1. Could the authors provide more mathematical details about the **delay embedding** implementation (dimensionality, architecture, initialization)?
2. How does Dedelayed behave when the **delay estimate τ** is inaccurate or unavailable at runtime?
3. Can the authors compare against other **delay-compensation** or **prediction-based** baselines (e.g., optical-flow warping, temporal extrapolation)?
4. What is the quantitative impact of using **pseudo-labels** instead of ground truth? Have you verified correlation with human annotations?
5. Is the fusion at a fixed early layer optimal? Would deeper or adaptive fusion yield different trade-offs?

---

### Official Review · Reviewer_cQuo · 2025-10-31

**Soundness:** 2
**Presentation:** 2
**Contribution:** 2
**Rating:** 4
**Confidence:** 3

**Summary:**

Existing frameworks of prediction models require large resources or are sensitive to the round-trip latency. This paper introduces Dedelayed, a framework for delay-compensated split inference that enables real-time visual prediction under communication latency. They design a method that fuses the features of a heavy-weight remote predictive model and a lightweight local model to enhance the model’s resistance to the round-trip delay.

**Strengths:**

- The work focuses on a relevant and practical issue in autonomous driving: handling communication latency during inference.
- The paper is clearly presented, and the figures are well designed and effectively support the explanations.

**Weaknesses:**

- The motivation section lacks sufficient detail and supporting experimental evidence. Although the introduction summarizes the weaknesses of existing solutions, it does not provide specific references to identify the related works. Moreover, key experimental results, such as resource consumption and sensitivity to round-trip latency in current approaches, are missing, which makes the motivation of this work less convincing.
-The manuscript would benefit from a more detailed description of the evaluation platform, including server configurations, embedded hardware specifications, and testing conditions. Providing this information would help justify the selection of models in this paper and improve the reproducibility of the results.
- Dedelayed lacks a detailed explanation of how the delay embedding is trained and integrated into the 3D ViT module, which is crucial for understanding how the method achieves performance gains under different and ever-changing latency conditions.
- Overall, the paper provides unclear motivation, vague design description, and incomplete evaluation details, which undermine confidence in the contribution.

**Questions:**

- The proposed method is mainly applied to segmentation tasks, and the tasks on autonomous driving are not restricted to segmentation. Could you justify the importance of raising the accuracy of segmentation from 50% to 65%?
- How sensitive is the proposed framework to the inference latency of the models deployed on each side? Since model latency directly affects the overall end-to-end delay, have you evaluated the framework using models with different computational costs to justify its robustness?

---

### Official Review · Reviewer_4R67 · 2025-10-31

**Soundness:** 3
**Presentation:** 2
**Contribution:** 1
**Rating:** 2
**Confidence:** 4

**Summary:**

Paper proposed a method to claims to mitigate arbitrary remote inference delays for cloud-assisted local image segmentation model, allowing the local device to produce low-latency outputs in real time using features from the heavy video-features based cloud model.

**Strengths:**

1. The paper tackles an important problem, as we are seeing an explosion of AR, mobile and household robotics, all of which need real-time guarantees but have limited compute and power. While the cloud has seemingly no such constraints, handling the latency and stochasticity of network conditions is a huge challenge when employed for these applications.
2. The idea of employing a heavy model in the cloud and a lightweight model on the edge, with careful feature-level coupling, is a good one - little research has been done on this specific aspect. Training such model pairs to be robust to network variability, feature staleness, asynchrony and optionality & resiliency (knowing when to use the cloud feature, when to ignore it etc), and supporting heterogeneous server-side models (say, a model family served with 1000 specialized LORAs each trained for specialized tasks) remain open challenges. Earlier methods (e.g., MARLIN, REACT) generally assumed that these models were pre-trained and focused primarily on late fusion of predictions.
3. The results, while in my preliminary, do show the advantage of such feature fusion design on the considered application.

**Weaknesses:**

1. The models and system architecture is not empirically justified. These design decisions while being crucial is not justified thoroughly, and it appears that there are obvious baselines and comparisons that should also be tested to justify the proposed method. Many questions remain unanswered, some example questions,

(a) Why fuse the remote features as a element wise addition? For e.g., given that both the models are transformers, would it not be better to pass tokens embeddings to the live model and employ these embedding as queries and perform cross-attention to fuse them with the local model's features. (i) One advantage with such a method is that we should be able to transmit K ($\lt \lt H \times W$) 1D latent tokens instead of transmitting image tokens (some factor of $H \times W$ tokens), ideally reducing turn-around time and managing latency better. (ii) The additional advantage of such a representation is that it allows us to additionally provide the delay as input to the local model as tokens and the delay is measured on the local device not on the remote device (because that's the clock we care about). I'm not sure why we should care about the delay measurement on the cloud.

(b) Why employ a video model remotely instead of a heavier image model (say a ViT-L or ViT-H)? What is the advantage of employing a 3D ViT on the server if the end goal is image-level segmentation, maybe previous image features are sufficient when fused correctly? -- This design decision is not justified.

2. The paper seems to cover only very simple models of network delay. Network emulation tools such as Mahimahi could be used to simulate more diverse network conditions (and ideally, user loads too). It is also unclear whether asynchrony is modeled, and if the approach is robust when features are received asynchronously. From the implementation section, it appears that specific fixed network delays were assumed, which is unrealistic, as both WiFi and LTE exhibit stochastic delay patterns. How do we train the local model to be robust these considerations is not obvious to me, and is an interesting research direction.
3. The paper is not carefully positioned with existing literature. Related work misses a lot of literature from the Systems, Mobile and Embedded Computing communities (Sensys, Mobisys, Mobicom, Sigcomm, IoTDI etc), creating an impression that the general problem has not been studied earlier -- many works in those communities have studied the problem of integrating delayed predictions in cloud-edge offloading situations (E.g., Glimpse [Sensys'15], MARLIN [Sensys'19], EdgeAssist [Mobicom'19], REACT [IoTDI'23] among others). Also missing are papers which tackle the real-time perception problem in vision and machine learning community, e.g. Streaming Perception [ECCV'20] (advocates for a generalized streaming metric to better quantify real-timeness), StreamYOLO [CVPR'22] (advocates predicting the future frame's output directly), DAMO-StreamNet [IJCAI'23] among many others.

**Questions:**

Please clarify the issues mentioned in weaknesses. I'm generally positive about the problem, however, I feel the paper raises more unanswered questions than it solves. Moreover, the writing and the figures leave much to be desired and need to be improved before the paper is ready for a venue like ICLR.

---

### Official Review · Reviewer_LPkV · 2025-10-31

**Soundness:** 2
**Presentation:** 1
**Contribution:** 1
**Rating:** 2
**Confidence:** 4

**Summary:**

This paper introduces Dedelayed, a framework that attempts to address the problem of communication delays in real-world (streaming) video segmentation tasks, where staleness of inputs or delays of results can cause severe issues on the downstream task.
Dedelayed builds upon collaborative inference, where a lightweight model is hosted at the edge and operates real-time on smaller resolution samples, whereas a larger heavyweight model is hosted remotely and operates on higher resolution, but stale inputs. The outputs of both models are eventually fused (via element-wise addition) locally and the output is emitted.
Evaluation boasts higher accuracy and lower latency on semantic segmentation tasks on the BDD100k dataset, compared to the local and remote-only baselines.

**Strengths:**

* The paper tackles a real problem in computer vision (and remote inference more generally), that of delayed inputs and results due to the roundtrip time of communicating between two endpoints.
* The proposed solution is simple and, on the evaluated scenario, effective, working also with pretrained networks.
* The mixed resolution processing is tackling a real choke point in computational throughput at the edge.

**Weaknesses:**

* The novelty and technical depth of the paper is limited, as well as its coverage of related work [a-e].
* The evaluation only covers a single simulated task on a single dataset, without comparing to other SOTA baselines .
    - I would suggest that the authors expand their evaluation setup to cover at least one additional dataset and set of models, along with potentially tying the model sizes to specific computational tiers of devices.
    - Including realistic fluctuations of bandwidth in mobile settings would also help showcasing the adaptability of the system.
* The proposed fusion model, i.e. element-wise addition of intermediate state, is quite simplistic and works on a limited set of possible delays and potentially offers no fault-tolerance. Moreover, the method mentions a learnable delay embedding, but does not fully describe its structure or training dynamics.
    - I would suggest a more rigorous and in-depth study of the possible fusion schemes and their dynamics.



[a] W. Nam, S. Lee, J. Lee, H. Choe, S. Ha and K. Lee, "N-Epitomizer: A Semantic Offloading Framework Leveraging Essential Information for Timely Neural Network Inferences," in IEEE Transactions on Networking, vol. 33, no. 3, pp. 1041-1055, June 2025, doi: 10.1109/TON.2024.3523891.
[b] Mingxuan Yan, Yi Wang, Xuedou Xiao, Zhiqing Luo, Jianhua He, and Wei Wang. 2023. Think before You Leap: Content-Aware Low-Cost Edge-Assisted Video Semantic Segmentation. In Proceedings of the 31st ACM International Conference on Multimedia (MM '23). Association for Computing Machinery, New York, NY, USA, 9224–9233. https://doi.org/10.1145/3581783.3613808
[c] Tianen Liu, Shuai Wang, Zheng Dong, Borui Li, and Tian He. 2025. From Perception to Computation: Revisiting Delay Optimization for Connected Autonomous Vehicles. ACM Comput. Surv. 57, 8, Article 200 (August 2025), 45 pages. https://doi.org/10.1145/3718361
[d] Y. Xie, Y. Guo, Z. Mi, Y. Yang and M. S. Obaidat, "Edge-Assisted Real-Time Instance Segmentation for Resource-Limited IoT Devices," in IEEE Internet of Things Journal, vol. 10, no. 1, pp. 473-485, 1 Jan.1, 2023, doi: 10.1109/JIOT.2022.3199921.
[e] X. Xiao, Y. Zuo, M. Yan, W. Wang, J. He and Q. Zhang, "Task-Oriented Video Compressive Streaming for Real-Time Semantic Segmentation," in IEEE Transactions on Mobile Computing, vol. 23, no. 12, pp. 14396-14413, Dec. 2024, doi: 10.1109/TMC.2024.3446185.

**Questions:**

* What are the requirements for alignment of the two models residing on the edge and cloud?
    - Do they need to be pretrained on the same dataset?
    - Do they need to share specific architectural components (e.g., depth, resolution of intermediate states for fusion)
* How does the current system deal with clients of different computational capacity? Does a separate model need to be trained for that particular tier of edge devices?
* What happens to the system if the delayed response from the server side does not arrive at all? Does the system provide any fault tolerance?
* What is the maximal delay (or feature staleness) that the models can handle? What does it depend on?
* What is the tradeoff between resolution and network capacity under a particular computational throughput?
* Could the two models share the same architecture and communicate intermediary state instead of raw inputs, like [f,g].


[f] Stefanos Laskaridis, Stylianos I. Venieris, Mario Almeida, Ilias Leontiadis, and Nicholas D. Lane. 2020. SPINN: synergistic progressive inference of neural networks over device and cloud. In Proceedings of the 26th Annual International Conference on Mobile Computing and Networking (MobiCom '20). Association for Computing Machinery, New York, NY, USA, Article 37, 1–15. https://doi.org/10.1145/3372224.3419194
[g] Kouris, A., Venieris, S.I., Laskaridis, S., Lane, N. (2022). Multi-Exit Semantic Segmentation Networks. In: Avidan, S., Brostow, G., Cissé, M., Farinella, G.M., Hassner, T. (eds) Computer Vision – ECCV 2022. ECCV 2022. Lecture Notes in Computer Science, vol 13681. Springer, Cham. https://doi.org/10.1007/978-3-031-19803-8_20

---

### Author Response · Authors · 2025-11-14

We thank each of the reviewers for their time and thoughtful feedback.

We acknowledge that our previous submission was incomplete and missing important details.

For instance, we omitted a detailed coverage of prior works. We have updated our manuscript following reviewer suggestions to include these more completely.

We are working to more clearly describe and justify all parts of the design, including those that the reviewers had questions about, and answer these questions directly in the main text.

We are withdrawing this version of the manuscript from consideration, and we thank the reviewers. Your feedback has materially strengthened our work.

---

### Note · Authors · 2025-11-14

I have read and agree with the venue's withdrawal policy on behalf of myself and my co-authors.